# A Simple Ratiometric Electrochemical Aptasensor Based on the Thionine–Graphene Nanocomposite for Ultrasensitive Detection of Aflatoxin B2 in Peanut and Peanut Oil

Fan Jia, Yuye Li, Qingfa Gong, Dong Liu *, Shuyun Meng, Chengxi Zhu  and Tianyan You *

Key Laboratory of Modern Agricultural Equipment and Technology, Ministry of Education, School of Agricultural Engineering, Jiangsu University, Zhenjiang 212013, China; ujsjf@outlok.com (F.J.); 2112016014@stmail.ujs.edu.cn (Y.L.); 2222116055@stmail.ujs.edu.cn (Q.G.); 2112116010@stmail.ujs.edu.cn (S.M.); dxzcx@jsut.edu.cn (C.Z.)
* Correspondence: dongliu@ujs.edu.cn (D.L.); youty@ujs.edu.cn (T.Y.)

**Abstract:** The accurate and reliable analysis of aflatoxin B2 (AFB2) is widely required in food and agricultural industries. In the present work, we report the first use of a ratiometric electrochemical aptasensor for AFB2 detection with high selectivity and reliability. The working principle relies on the conformation change of the aptamer induced by its specific recognition of AFB2 to vary the ratiometric signal. Based on this principle, the proposed aptasensor collects currents generated by thionine–graphene composites ($I_{THI}$) and ferrocene-labeled aptamers ($I_{Fc}$) to output the ratiometric signal of $I_{THI}/I_{Fc}$. In analysis, the value of $I_{THI}$ remained stable while that of $I_{Fc}$ increased with higher AFB2 concentration, thus offering a "signal-off" aptasensor by using $I_{THI}/I_{Fc}$ as a yardstick. The fabricated aptasensor showed a linear range of 0.001–10 ng mL$^{-1}$ with a detection limit of 0.19 pg mL$^{-1}$ for AFB2 detection. Furthermore, its applicability was validated by using it to detect AFB2 in peanut and peanut oil samples with high rates of recovery. The developed ratiometric aptasensor shows the merits of simple fabrication and high accuracy, and it can be extended to detect other mycotoxins in agricultural products.

**Keywords:** ratiometry; electrochemical aptasensor; thionine; aflatoxin B2

## 1. Introduction

Aflatoxins (AFs) are difuran ring toxoids generated by strains such as Aspergillus flavus and Aspergillus parasiticus [1]. With regard to their hepatotoxicity and carcinogenicity for humans, the International Agency for Research on Cancer has classified AFs as Group I human carcinogens [2,3]. Many countries and world organizations have imposed strict conditions on AFs; e.g., the maximum residue limit of AFs in foods is 15 ng mL$^{-1}$ according to the International Codex Alimentarius Commission (ICAC) [4,5]. Aflatoxin B2 (AFB2) in particular has been confirmed to be extremely hazardous [6,7]. Many state-of-the-art detection technologies have been developed for the detection of AFB2, such as the enzyme-linked immunosorbent assay (ELISA) and high-performance liquid chromatography coupled with fluorescence detection (HPLC-FLD) [8–10].

Besides the abovementioned techniques, electrochemical methods have attracted enormous attention for the advantages of simplicity, rapid response and low cost [11–15]. Various electrochemical sensors have been constructed for the sensitive analysis of AFs, such as the multifunctional DNA nanotube-based electrochemical sensor and the dual-mode sensor using a combination of differential pulse voltammetry and electrochemical impedance spectroscopy [16,17]. Nevertheless, there had been relatively few reports focused on the electrochemical sensing of AFB2 with excellent selectivity until the successful screening of the AFB2 aptamer in 2015 [18]. Taking advantage of the high specificity of the aptamer, Lu and co-workers construct a colorimetric aptasensor based on sodium chloride-induced aggregation of AuNPs for visualization detection of AFB2 [19]. A screen-printed

electrode-based aptasensor has been reported that employs the aptamer to significantly enhance the sensitivity and selectivity in AFB2 analysis [20]. Notably, the analytical properties of electrochemical aptasensors with a single-signal detection mode can be influenced by environmental factors. In this regard, ratiometry has been introduced into electrochemical sensing with the aim of acquiring better reliability and reproducibility [21,22]. Typically, its working principle relies on the utilization of intrinsic built-in correction, reading two independent electrochemical signals of probes at different potentials. Ferrocene (Fc) and methylene blue (MB) are the best among the probes frequently used to fabricate ratiometric electrochemical aptasensors [23–25]. Generally, probes are used to construct ratiometric sensors in the following ways: (1) by using probe-labeled DNA, in which the target triggers a conformational change in the DNA strand and the electrochemical signal of the probe also changes accordingly [24]; or (2) by using double-strand DNA (dsDNA) that can specifically absorb probes. There are several redox probes that can be specifically adsorbed by dsDNA. For example, the target that triggers or hinders DNA self-assembly can vary the amount of dsDNA to adsorb MB, which significantly changes the electrochemical signal of the MB [25]. Meanwhile, several novel beacon molecules have also been employed to construct ratiometric sensors, such as copper ions ($Cu^{2+}$). Using Fc as a reference signal, the reduction peak current of $Cu^{2+}$ was used as a response signal to achieve the ratiometric detection of $Cu^{2+}$ in a rat brain [26]. Yang et al. described a metal–organic framework (MOF)-based ratiometric sensor based on the ion-exchange reaction of $Cu^{2+}$ with target ions ($Pb^{2+}$, $Cd^{2+}$) [27]. The presence of the target ion led to a decrease in $Cu^{2+}$ in MOFs, and then the reduction current of $Cu^{2+}$ ($I_{Cu2+}$) fell while that of $Pb^{2+}$ ($I_{Pb2+}$) or $Cd^{2+}$ ($I_{Cd2+}$) increased. In this way, the ratiometric detection was realized by using $I_{P2+}/I_{Cu2+}$ or $I_{Cd2+}/I_{Cu2+}$ as the readout signal. Notably, it is difficult for many electroactive species, such as $Cu^{2+}$, to be labeled or absorbed in DNA with high stability to output a reference signal. However, the investigations into such species may provide alternative ways to construct ratiometric sensors. Among these species, thionine (THI) has been widely explored for its potential to provide internal calibration in electrochemical ratiometric sensors [28,29].

Inspired by these studies, our group has developed a ratiometric aptasensor using THI-loaded reduced graphene oxide (THI-rGO) as a substrate [30]. The disassociation of the Fc-tagged aptamer with complementary DNA in the presence of the target produces a reduced response current from Fc, offering a "turn-off" readout mode. In this study, in order to improve the analytical performance, we further developed a "turn-on" ratiometric electrochemical aptasensor via a relatively simple route for the sensitive detection of AFB2 (Scheme 1). The THI-rGO composite was electrostatically adsorbed on the surface of a glassy carbon electrode (GCE) and used as a substrate to output a reference signal, while the Fc-tagged aptamer was assembled on THI-rGO through electrostatic interaction [31]. In the analysis, the binding of AFB2 with the aptamer made Fc approach GCE. Consequently, $I_{Fc}$ increased at higher AFB2 concentrations, while $I_{THI}$ remained constant. This aptasensing assay using $I_{THI}/I_{Fc}$ as a ratiometric signal showed high selectivity and sensitivity in AFB2 analysis.

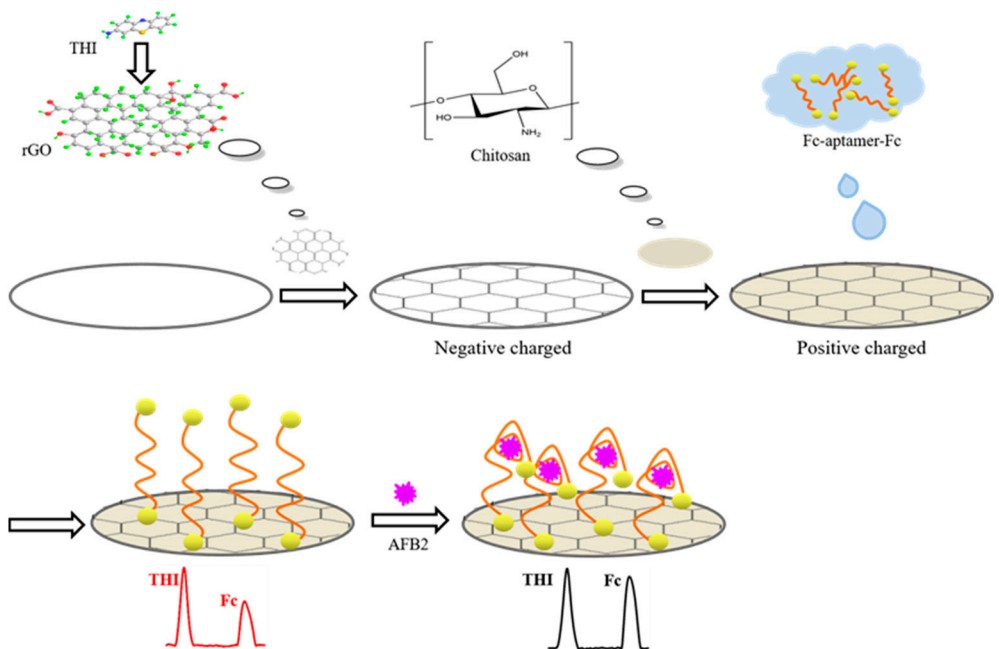

**Scheme 1.** Schematic illustration of the fabrication of a ratiometric aptasensor for AFB2 detection.

## 2. Materials and Methods

### 2.1. Reagents and Materials

Sodium hydrogen phosphate ($Na_2HPO_4$) and sodium dihydrogen phosphate ($NaH_2PO_4$) were purchased from Sinopharm Chemical Reagent (Shanghai, China). Fumonectin B1 (FB1, 96.0%, Shanghai Aladdin Bio-Chem Technology Co., Ltd., Shanghai, China), Chitosan (CS, Aladdin), reduced graphene oxide (rGO, XFNANO Inc., Nanjing, China), thionine (THI, Alfa Aesar, Haverhill, MA, USA), aflatoxin B1 (AFB1, 98.0%, Acros Organics, NJ, USA), aflatoxin M1 (AFM1, 98.0%, Acros Organics), and aflatoxin B2 (AFB2, 98.0%, Acros Organics) were employed for experiments. The aptamer of AFB2 with 81 bases was synthesized by Sangon Biotech Co., Ltd. (Shanghai, China), and its DNA sequence was as follows [18]:

Fc-Aptamer-Fc: 5′-Fc-AGCAGCACAGAGGTCAGATGCTGACACCCTGGACCTT GGGATTCCGGAAGTTTTCCGGTACCTATGCGTGCTACCGTGAA-Fc-3′

The aptamer (100 μM) was stored in 10 mM Tris-HCl buffer (pH 7.4) containing 1 mM KCl. The AFB2 was dissolved with methanol and diluted with phosphate-buffered saline (PBS, pH = 7.0) to obtain stock solutions with different concentrations.

### 2.2. Apparatus

A UV-vis/NIR spectrophotometer (Shimadzu UV-3600plus, Shimadzu, Nakagyo-ku, Kyoto, Japan) was used to obtain UV-vis absorption spectra. X-ray photoelectron spectroscopy (XPS) analysis was performed with an ESCALAB250 X-ray photoelectron spectrometer with the following X-ray source: monochromated Al Kα; hν = 1451 eV; spot size: 650 μm (Thermo Fisher Scientific, Waltham, MA, USA). Raman spectra were registered in LabRam HR800 using a 633 nm laser module and 5 s exposure time (Horiba Scientific, Longjumeau, France). Scanning electron microscope (SEM) images were obtained with a SU8020 SEM with acceleration voltages of 3.0 kV (Hitachi, Tokyo, Japan).

Alternating current voltammetry (ACV) and electrochemical impedance spectroscopy (EIS) were evaluated on a CHI 852D electrochemical workstation (Shanghai Chenhua, Shanghai, China) and a Zennium electrochemical workstation (Zennium, Zahner, Germany), respectively. EIS measurements were performed in 0.1 M KCl solution containing 5 mM $[Fe(CN)_6]^{3-/4-}$ with a frequency range from 10 kHz to 0.1 Hz. ACVs were carried out over a potential range from −0.4–0.7 V in 0.1 M PBS (pH 7.0), with a frequency of 25 Hz, a step potential of 4 mV, and an amplitude of 25 mV. This work employed a common three-electrode system for measurement, using a platinum wire as the counter electrode, a

Ag/AgCl (sat. KCl) electrode as the reference electrode and a glassy carbon electrode as the working electrode.

### 2.3. Synthesis of THI-rGO

The THI-rGO nanocomposite was prepared according to a previously reported method with some appropriate modifications [32]. Briefly, 3 mL of THI aqueous solution (5 µg mL$^{-1}$) was added into 15 mL of rGO aqueous dispersion (0.1 mg mL$^{-1}$) and stirred for 12 h. Then, the dispersion was centrifuged and washed before being dispersed in Milli-Q grade water.

### 2.4. Fabrication of the Aptasensor

The GCE was polished with 0.05 µm alumina powder to acquire a mirror-like surface and then rinsed with ethanol and water for 30 s. Then, 6 µL of THI-rGO (1 mg mL$^{-1}$) was dispersed, following which 8 µL of CS (5 wt%, pH 5.0) solution was cast on an electrode and then dried at 37 °C. Next, the electrode was rinsed with 10 mM PBS and incubated with 6 µL 2.4 µM Fc-Aptamer-Fc (Fc-Apt-Fc) at room temperature for 12 h, then stored at 4 °C before use.

### 2.5. Analysis of AFB2

Six microliters of AFB2 solution was cast on the aptasensor with different concentrations. After incubation for 100 min at room temperature, the aptasensor was washed with PBS before measurements.

### 2.6. Real Sample Pretreatment

The peanut and peanut oil samples were obtained from the local supermarket (Zhenjiang). The extraction process was performed according to the method reported by Deng and Hou with some modifications [33,34]. The extraction solution was prepared by mixing 14 mL methanol and 6 mL ultrapure water. The mixture of the extraction solution and the 5 g samples was shaken for 60 min and then centrifuged at 6000 rpm for 10 min to extract the supernatant. The resulting solution was then dialyzed with an ultrafiltration membrane (0.22 µm). Then, 10 mL of the treated sample was fortified by adding AFB2 standard solutions to prepare the spiked samples containing 10, 1 and 0.01 ng mL$^{-1}$ of AFB2.

The real samples were analyzed with an Agilent 1290 Infinity UHPLC coupled to a 6460 Triple Quad mass spectrometer from ICAS Testing Technology Service Co., Ltd. (Shanghai, China).

### 3. Results and Discussion

#### 3.1. Characterization of THI-rGO Nanocomposite

The morphology and elemental composition of the THI-rGO nanocomposite were investigated. Both rGO (Figure 1A) and THI-rGO (Figure 1B) showed flake-like shapes with wrinkles, but nitrogen and sulfur elements were only observed in THI-rGO (Figure 1C), suggesting the presence of THI [35]. As shown in Figure S1, X-ray diffraction of rGO revealed a diffraction peak at 23.6° indexed to the (002) plane of rGO. Following Bragg's law, the interplanar distance of rGO was calculated to be 0.37 nm due to the presence of oxygen-containing groups. UV-vis spectrometry was used to characterize the interactions of THI, rGO, and THI-rGO. In Figure 1D, rGO (curve a) shows a strong absorption peak at about 270 nm, assigned to the reduced GO. The characteristic absorption peaks of THI-rGO (curve b) and THI (curve c) at 283 nm, 564 nm and 600 nm were attributed to the π–π* transition of aromatic rings, H-type dimer aggregate, and the n–π* transitions of C=N, respectively [36]. UV-vis spectra showed that the composite material had the characteristic peaks of thionine and rGO, which proved that THI molecules and rGO existed in the composite material [37]. To further investigate the combination method of the composite material, the Raman spectra characteristic absorption bands of rGO (Figure 1E, curve a), THI-rGO (Figure 1E, curve b), and THI (Figure 1E, curve c) were compared. Here, the characteristic peaks at 1152 and 1189 cm$^{-1}$, 1387 and 1428 cm$^{-1}$,

$1504$ cm$^{-1}$, and $1627$ cm$^{-1}$ were attributed to C–H, C–N and NH$_2$ functional groups and the C=C bond in the THI molecule [38]. Furthermore, the characteristic G band attributed to graphene ($1583$ cm$^{-1}$) was consistent with the previous results from the literature, which indicates that THI functionalization does not introduce lattice defects on rGO, confirming the noncovalent interaction hypothesis [39]. Moreover, Zeta potential measurement was used to investigate the surface charge of the stacked structures. The reduced Zeta potential of THI-rGO ($-6.81$ mV) as compared with that of rGO ($-18.5$ mV) indicated the presence of electrostatic interaction (Figure 1F) [40,41].

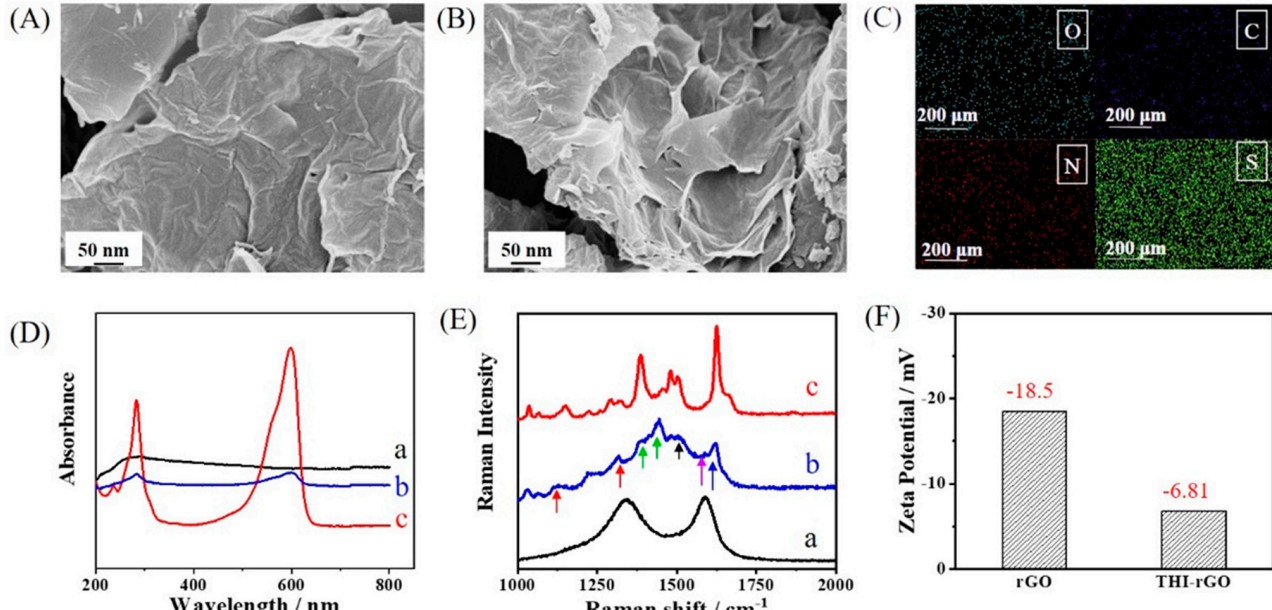

**Figure 1.** SEM characterization of (**A**) rGO and (**B**) THI-rGO. (**C**) Mapping images of THI-rGO. (**D**) UV-vis absorption spectra and (**E**) Raman spectra of (a) rGO, (b) THI-rGO and (c) THI. (**F**) Zeta potential of rGO and THI-rGO.

Subsequently, X-ray photoelectron spectroscopy (XPS) analysis was used to further verify the noncovalent boding of THI and rGO. In Figure 2A, the XPS spectrum of THI-rGO can be seen with peaks at $532.7$, $400.1$, $285.4$, and $165.3$ eV associated with O$_{1s}$, N$_{1s}$, C$_{1s}$, and S$_{2p}$, respectively [42]. However, no peaks assigned to N$_{1s}$ and S$_{2p}$ were observed in the XPS spectrum of rGO (Figure 2B). The atomic ratio of C to O in THI-rGO was significantly reduced compared with that of rGO, which was attributed to the oxygen in THI. The N$_{1s}$ deconvolution spectrum of THI-rGO (Figure 2C) revealed several chemical bonds for nitrogen: C–N=C ($399.3$ eV), C–NH$_2$ ($400.3$ eV), and NH$_2$ ($403.2$ eV). The S$_{2p}$ deconvolution spectrum (Figure 2D) displayed two components for sulfur bonds: S$_{2p3/2}$ C–S–C ($168.7$ eV) and S$_{2p1/2}$ C–S–C ($165.2$ eV) [43,44].

These results verified the successful preparation of the THI-rGO composite via the electrostatic assembly.

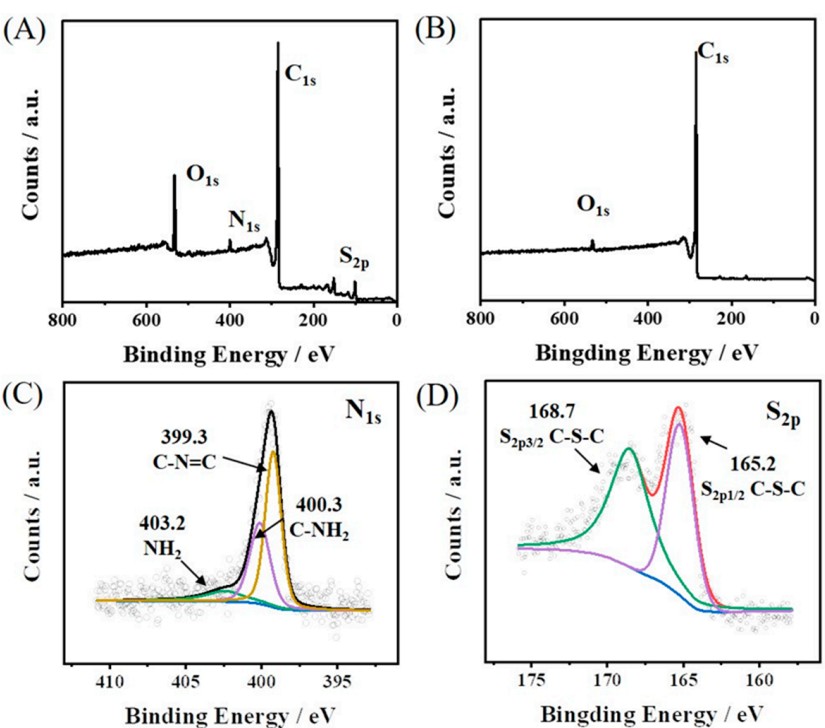

**Figure 2.** Survey XPS spectra of (**A**) THI-rGO and (**B**) rGO. High-resolution (**C**) $N_{1s}$ and (**D**) $S_{2p}$ XPS spectra of THI-rGO.

### 3.2. Feasibility of the Ratiometric Electrochemical Aptasensor for AFB2 Assay

The feasibility of the aptasensor was assessed using alternating current voltammetry (ACV) (Figure 3A). After the modification of the THI-rGO, a remarkable peak located at −0.2 V was observed, which was ascribed to the oxidation of THI (Figure 3A, curve a). Chitosan, as a carrier for electrostatic adsorption, did not significantly reduce the current intensity of the THI (curve b). The modification of the Fc-Aptamer-Fc (Fc-Apt-Fc) led to the emergence of a characteristic peak of Fc at 0.5 V (curve c). Upon the addition of AFB2, the peak current of the THI ($I_{THI}$) showed no obvious change, while the current of the Fc ($I_{Fc}$) significantly increased (curve d).

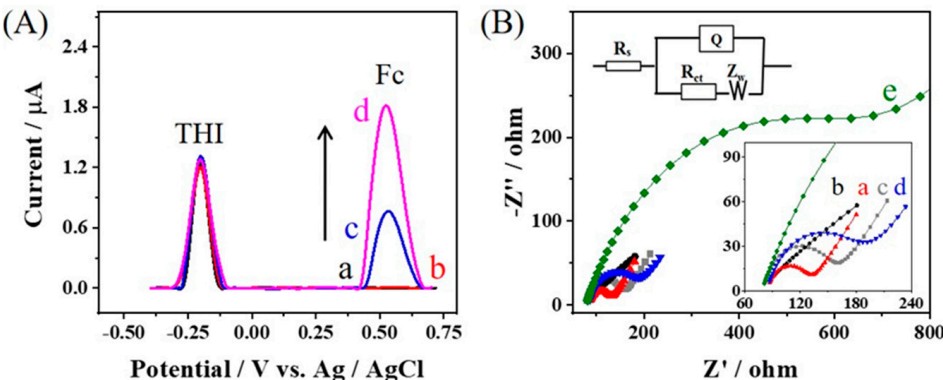

**Figure 3.** (**A**) ACV characterization of (a) THI-rGO/GCE, (b) CS/THI-rGO/GCE, (c) Fc-Apt-Fc/CS/THI-rGO/GCE, and (d) AFB2 (1 ng mL$^{-1}$)/Fc-Apt-Fc/CS/THI-rGO/GCE. (**B**) Nyquist plots for different electrodes: (a) GCE, (b) THI-rGO/GCE, (c) CS/THI-rGO/GCE, (d) Fc-Apt-Fc/CS/THI-rGO/GCE, and (e) AFB2 (1 ng mL$^{-1}$)/Fc-Apt-Fc/CS/THI-rGO/GCE. Inset is the Randles equivalent circuit used to fit the impedance spectra.

Figure 3B shows the electrochemical impedance spectra for different electrodes. Nyquist plots were recorded to characterize the state of the electrode surface during the process of

constructing the aptasensor. Electronic transfer resistance ($R_{ct}$) was used to estimate the interface properties. As shown in Figure 3B, $R_{ct}$ reduced from 76.6 $\Omega$ at GCE (curve a) to 18.4 $\Omega$ at GCE/THI-rGO (curve b) after the modification of the THI-rGO [45]. After the introduction of chitosan (CS), $R_{ct}$ increased to 45.8 $\Omega$ (curve c). The presence of an aptamer with poor conductivity led to an increase of $R_{ct}$ (103 $\Omega$) (curve d) [46]. Upon the addition of AFB2, $R_{ct}$ showed an obvious increase to 660 $\Omega$ (curve e).

Thus, the presence of AFB2 led to an increase in $I_{Fc}$ and $R_{ct}$, and this observation was ascribed to the reduced distance between Fc and the electrode resulting from the formation of the AFB2–Apt complex [47]. CS played a vital role in connecting and stabilizing the aptamer [48,49]. The strong electrostatic adsorption between CS and the aptamer did not change the spatial conformation of the aptamer, but the AFB2–Apt conjugate can be retained on the electrode surface [50]. The above explanation was supported by the increased $R_{ct}$ after adding AFB2. Thus, such a ratiometric strategy can be used for AFB2 detection.

### 3.3. Optimization of Experimental Conditions

Experimental conditions including the incubation time of the target, the aptamer concentration, and the incubation time of the aptamer were optimized to improve the analytical performance of the aptasensor.

As shown in Figure S2A, $I_{Fc}$ increased until the concentration of the aptamer ($C_{Aptamer}$) reached 2.0 $\mu$M, and then it remained stable due to the saturation of the aptamer on the electrode. The value of 2.4 $\mu$M was thus chosen for $C_{Aptamer}$.

Next, the effect of the aptamer incubation time on $I_{Fc}$ was investigated. The Fc-Apt-Fc was incubated at 4 °C. The amount of Fc on the electrode reached saturation after 10 h incubation (Figure S2B). Therefore, 12 h was selected as the optimal incubation time.

The time for the reaction between the aptamer and AFB2 was optimized. As shown in Figure S2C, $I_{Fc}$ reached a maximum at 80 min, suggesting the completely binding of AFB2 with the aptamer. Thus, 100 min was selected as the optimal reaction time.

### 3.4. Determination of AFB2

With the optimal parameters, it can be seen from Figure 4A, showing ACVs with multiple concentrations of AFB2 ($C_{AFB2}$), that the value of $I_{Fc}$ increased with higher $C_{AFB2}$ while that of $I_{THI}$ remained constant. The ratiometric signals of $I_{THI}/I_{Fc}$ obtained were then plotted against the logarithm of $C_{AFB2}$ (Figure 4B). According to the constructed calibration plot, a linear range of 0.001–10 ng mL$^{-1}$ was obtained with a regression equation of $I_{THI}/I_{Fc}$ = $-0.0883$ log$C_{AFB2}$ $-$ 0.413 ($R^2$ = 0.995). The limit of detection (LOD) was estimated to be 0.19 pg mL$^{-1}$ (3SD/slope; SD is the standard deviation of 10 blank samples). Figure 4C shows the calibration curves using individual $I_{Fc}$ as the response signal. A linear relationship was obtained in the ranges from 3.3 pg mL$^{-1}$ to 1 ng mL$^{-1}$ with a regression equation of $I_{Fc}$ = 0.293 log$C_{AFB2}$ + 5.39 ($R^2$ = 0.994). The linear range of the ratiometric aptasensor was better compared to the single-signal aptasensor. Thus, as compared with the single-signal mode, the ratiometric strategy with a built-in reference signal allows a wider linear range for the detection of AFB2. The lamellar structure of rGO has a larger specific surface area, which gives it stronger load capacity. Therefore, the built-in reference signal provided by the THI–rGO composite enhances the stability of the ratiometric aptasensor.

Subsequently, the analytical performance of the ratiometric electrochemical aptasensor was compared with previous reports. As shown in Table 1, compared with HPLC-FLD, pCEC-FLD, and immunochromatography, the linear range and detection limit of the developed aptasensor were reduced by at least two orders of magnitude [51–55]. Although previously reported electrochemical sensors offered relatively wider linear ranges, the developed ratiometric aptasensor has enhanced accuracy [20].

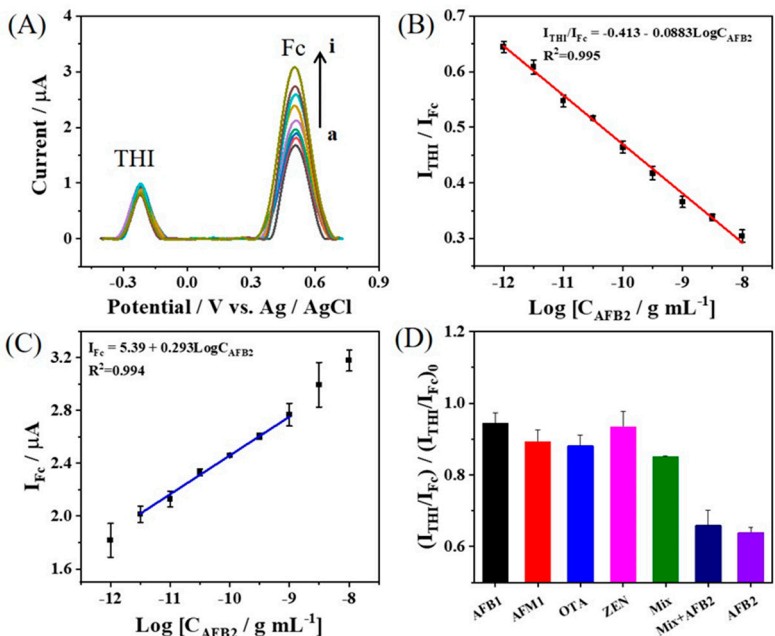

**Figure 4.** (**A**) ACV responses of the aptasensor to AFB2 with a concentration of (a–i): 0.001, 0.003, 0.01, 0.03, 0.1, 0.3, 1, 3, and 10 ng mL$^{-1}$. Calibration plot of (**B**) $I_{THI}/I_{Fc}$ and (**C**) $I_{Fc}$ vs. the logarithm of the $C_{AFB2}$. (**D**) Comparisons of the aptasensor responses to 0.01 ng mL$^{-1}$ solution of AFB2 and 0.2 ng mL$^{-1}$ solution of AFB1, AFM1, OTA, ZEN, and the mixture of the interferences.

**Table 1.** Comparisons of the developed aptasensor with several previously reported methods for AFB2 detection.

| Method | Linear Range (ng mL$^{-1}$) | Detection Limit (ng mL$^{-1}$) | Ref. |
|---|---|---|---|
| HPLC-FLD [1] | 0.2–7.0 | 0.1 | [51] |
| HPLC-FLD [1] | 1.74–17.4 | 0.66 | [52] |
| Immunochromatography | 0.5–2.5 | 0.16 | [53] |
| pCEC-FLD [2] | $10^2$–$3 \times 10^3$ | 10 | [54] |
| UV-vis | $2.5 \times 10^{-2}$–10 | $8.3 \times 10^{-3}$ | [19] |
| Electrochemistry | $10^{-4}$–$10^2$ | $10^{-4}$ | [20] |
| Electrochemistry | $10^{-7}$–$10^{-3}$ | $2 \times 10^{-7}$ and $6 \times 10^{-7}$ | [55] |
| **Ratiometric electrochemistry** | **$10^{-3}$–10** | **$1.9 \times 10^{-4}$** | **This work** |

[1] HPLC-FLD: high performance liquid chromatography coupled to a fluorescence detector; [2] pCEC-FLD: pressurized capillary electrochromatography laser-induced fluorescence detection.

### 3.5. Selectivity, Reproducibility, and Stability of Proposed Aptasensor

Selectivity is of extreme importance for sensors. In this work, the responses of the developed aptasensor to common mycotoxins, including AFB1, AFM1, OTA, ZEN, and their mixture, were estimated. The concentration of each interferent (0.2 ng mL$^{-1}$) was 20 times higher than that of AFB2 (0.01 ng mL$^{-1}$). As shown in Figure 4D, a relatively low signal response was observed in the presence of all these interferent mycotoxins except AFB2 and the developed aptasensor showed high specificity for AFB2, thereby demonstrating its potential applications in real sample analysis.

Then, reproducibility testing was performed using six different aptasensors, and the relative standard deviation (RSD) was 4.2% (Figure S3A). To estimate the stability, a batch of developed aptasensors were stored at 4 °C, and the stability test was carried out every other day. As shown in Figure S3B, the intensity of $I_{THI}/I_{Fc}$ decreased as the storage time

extended and the response rate of the aptasensor remained 92.4% of that of the first day on the seventh day, showing high stability.

### 3.6. Analysis of AFB2 in Peanut Powder and Peanut Oil

To assess the practical applicability, the developed ratiometric electrochemical aptasensor was applied to AFB2 analysis in peanut and peanut oil samples (0.01, 1, and 10 ng mL$^{-1}$). In order to verify the accuracy of the as-developed method, these samples were also analyzed using HPLC-MS/MS. As shown in Table 2, the recovery rates with the developed aptasensor were in the range from 98.6% to 108%, with RSD below 9.1%, which was consistent with the results from HPLC-MS/MS. All the above results suggest that the developed aptasensor can be used for practical applications.

**Table 2.** Determination of AFB2 in peanut and peanut oil with the proposed aptasensor (n = 3) and HPLC-MS/MS.

| Sample | Spiked (ng mL$^{-1}$) | Developed Aptasensor | | | HPLC-MS/MS |
|---|---|---|---|---|---|
| | | Detected (ng mL$^{-1}$) | Recovery (%) | RSD (%) | Detected (ng mL$^{-1}$) |
| Peanut | 0.0100 | 0.0108 | 108 | 6.9 | - [a] |
| | 1.00 | 0.986 | 98.6 | 6.2 | 0.997 |
| | 10.0 | 10.4 | 104 | 6.1 | 10.8 |
| Peanut oil | 0.0100 | 0.0108 | 108 | 9.1 | - [a] |
| | 1.00 | 1.07 | 107 | 5.3 | 0.903 |
| | 10.0 | 10.4 | 104 | 5.5 | 10.4 |

[a] Not detected.

### 4. Conclusions

A simple ratiometric electrochemical aptasensor was constructed to detect AFB2. The THI-rGO nanocomposite was prepared using electrostatic interaction and bonded to the electrode surface via electrostatic adsorption to the output reference signal, while a Fc-labeled aptamer was used to generate the response signal. Raman spectroscopy, XPS, SEM, and Zeta potential measurements were conducted to prove the successful synthesis of materials. The proposed aptasensor demonstrated a wide linear range with a relatively low detection limit for AFB2 analysis. The practical applicability was validated using peanut and peanut oil samples analysis with the developed sensor. Moreover, the aptasensor displayed results consistent with HPLC-MS/MS. This aptasensor offers a simple but reliable method for the detection of mycotoxins in agricultural production.

**Supplementary Materials:** The following supporting information can be downloaded at: https://www.mdpi.com/article/10.3390/chemosensors10050154/s1, Figure S1: XRD spectra of rGO, Figure S2: Optimization of experimental conditions: (A) effect of aptamer concentration, (B) incubation time of the aptamer, (C) incubation time of AFB2 for the peak current. Error bars represent the relative standard deviation (RSD, n = 3), Figure S3: (A) The reproducibility for AFB2 detection from six parallel measurements; (B) the sensing results of the proposed aptasensor for AFB2 detection after storage for 1–7 days.

**Author Contributions:** F.J.: Investigation, Data curation, Writing—original draft. Y.L. and Q.G.: Resources, Software, Writing—review & editing. D.L.: Methodology, Conceptualization, Formal analysis, Writing—review & editing. S.M.: Resources, Software. C.Z.: Resources, Software. T.Y.: Supervision, Conceptualization, Writing—review & editing. All authors have read and agreed to the published version of the manuscript.

**Funding:** We would like to thank the support from the National Natural Science Foundation of China (No. 61901193, 22074055, 62001197), the Natural Science Foundation of Jiangsu Province (No. BK20200104), the Innovation/Entrepreneurship Program of Jiangsu Province, the Project of the Faculty of Agricultural Equipment of Jiangsu University, and the Priority Academic Program Development of Jiangsu Higher Education Institutions.

**Data Availability Statement:** The data presented in this study are available in the Supplementary Materials section.

**Conflicts of Interest:** The authors declare no conflict of interest.

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
