# Peer review of "A Simple Ratiometric Electrochemical Aptasensor Based on the Thionine–Graphene Nanocomposite for Ultrasensitive Detection of Aflatoxin B2 in Peanut and Peanut Oil"

_chemosensors, doi:10.3390/chemosensors10050154_

Round 1
Reviewer 1 Report
The present article shows a clever way to use ratiometric electrochemical aptasensor for AFB2 detection with high selectivity and reliability. The analytical parameters were well evaluated as the characterization techniques used. However, the authors need to discuss better the sensing mechanism of the proposed sensor.
Author Response
Thanks for your comments. Our response is provided in the attached file.

Reviewer 2 Report
In this paper, the authors explored and proposed a ratiometric electrochemical aptasensor based on thionine loaded reduced graphene oxide (THI-rGO) as substrate for detection of aflatoxin B2 (AFB2). In my opinion, the manuscript is worthy of publication in this journal, after clarifying the following questions:
- Some grammatical mistakes were made in this manuscript, please correct them.
- In Fig. 1C, the author should improve the contrast of the figures to get clearer experimental results. The current figures are too blurry.
- In Fig. 4B, I suggest that the y-coordinate be changed as “IFc/ITHI”. Changes are also required elsewhere in the manuscript.
- In Table 1, figures should be expressed in scientific notation, such as “0.0000001” should be change as “10-7”.
Author Response

(The authors gave the same response as above.)

Reviewer 3 Report
In this manuscript the authors report a ratiometric electrochemical method for aflatoxin B2 (AFB2) determination in peanut and peanut oil. The used aptasensor was prepared by modifying a GCE with thionine-graphene composite (ITHI) and ferrocene-labelled aptamer (IFc). The ratiometric ITHI/IFc signal increased with increasingAFB2 concentrations and was used for AFB2 quantification at ng/mL levels. The method is interesting and enables a very sensitive AFB2 determination. Some minor suggestions which may improve the manuscript before publication are listed below:
- Line 119: "3 mL THI aqueous solution (5 µg mL-1) was added into 15 mL rGO (0.1 mg mL-1) dispersion."
The authors wrote that THI was dissolved in water but which was the solvent for the rGO dispersion? - Line 204 "Figure 3B shows the electrochemical impedance spectroscopy"
Spectroscopy is the technique, Fig. 3B shows the electrochemical impedance spectra. - The aptamer concentration remained constant after 2.0 microM, the aptamer incubation time after 10 h and the target incubation time of 80 min, Why didn't chose the authors these values as optimum and they selected the aptamer concentration of 2.4 microM, aptamer incubation time 12 h and target incubation time of 100 min as optimum values for further investigations?
- Lines 233-235: "Under the optimal parameters, ACVs at multiple concentrations of AFB2 (CAFB2) were provided in Figure 4A that the value of ITHI increased with higher CAFB2 while that of IFc kept constant. "
Please revise the sentence. The ITHI remained constant and IFC increased with increasing AFB2 concentrations. - Which was the limit of detection for the method using the single-signal aptasensor?
- Why are in Table 1 two values for the detection limit corresponding to reference [55]?
Author Response

(The authors gave the same response as above.)
